# Numerical Analysis on Velocity and Temperature of the Fluid in a Blast Furnace Main Trough

**Yao Ge [1], Meng Li [1], Han Wei [1], Dong Liang [2], Xuebin Wang [2] and Yaowei Yu [1,*]**

[1] State Key Laboratory of Advanced Special Steel, Shanghai Key Laboratory of Advanced Ferrometallurgy, School of Materials Science and Engineering, Shanghai University, Shanghai 102100, China; ge_geyao@hotmail.com (Y.G.); L_limeng@163.com (M.L.); weihan@shu.edu.cn (H.W.)

[2] LaiSteel Research and Technology Center, LaiSteel, Jinan 250000, China; ironlaoliang@163.com (D.L.); erli2000@126.com (X.W.)

* Correspondence: yaowei.yu@hotmail.com

**Abstract:** The main trough is a part of the blast furnace process for hot metal and molten slag transportation from the tap hole to the torpedo, and mechanical erosion of the trough is an important reason for a short life of a campaign. This article employed OpenFoam code to numerically study and analyze velocity, temperature and wall shear stress of the fluids in the main trough during a full tapping process. In the code, a three-dimensional transient mass, momentum and energy conservation equations, including the standard k-ε turbulence model, were developed for the fluid in the trough. Temperature distribution in refractory is solved by the Fourier equation through conjugate heat transfer with the fluid in the trough. Change velocities of the fluid during the full tapping process are exactly described by a parabolic equation. The investigation results show that there are strong turbulences at the area of hot metal's falling position and the turbulences have influence on velocity, temperature and wall shear stress of the fluid. With the increase of the angle of the tap hole, the wall shear stress increases. Mechanical erosion of the trough has the smallest value and the campaign of the main trough is estimated to expand over 5 days at the tap hole angle of 7°.

**Keywords:** main trough; transient fluid of hot metal and molten slag; wall shear stress; conjugate heat transfer; refractory

---

## 1. Introduction

The main trough of the blast furnace is a drainage channel for molten iron and slag. In tapping of a 3000 m$^3$ blast furnace, 4 to 7 tons per minute of molten slag and hot metal with 1773 K flows into the main trough from a tap hole. Tapping time changes from 70 to 120 min and tapping number is around 15 every day [1]. Then, with gravity force, molten slag moves to a skimmer on the top of hot metal and is separated into a slag trough by the skimmer. Due to a harsh working environment, the main trough of the 3000 m$^3$ blast furnace has 9 to 10 campaigns per year and 45 tons of casting material (with a price of $857 per ton) is needed for every campaign. Each campaign runs for about 35 days and needs five minor maintenances. Each minor maintenance consumes 3 tons of ramming material (with a price of $823 per ton). The cost of the blast furnace main trough is around $0.52 million per year excluding manpower, time and environmental cost [2]. The maintenance cost of the main trough is very expensive. Therefore, the internal state of the main trough should be known, and the erosion mechanism of refractory materials must be understood by the operators and the managers of the blast furnace. Erosional factors of the blast furnace trough include [3]: (1) Mechanical (physical) erosion of fluid flows of molten slag and hot metal, (2) chemical reaction erosion between refractory and the fluid, and (3) thermal stress erosion of intermittent tapping. The main one is the first (mechanical erosion),

which is proven by the fact that the erosional extent of an iron storage trough is quite little in the new generation of huge blast furnaces.

In order to reduce the erosion of the main trough, scientists and engineers have done a lot of works to understand the inner situation of the main trough. There are two approaches to study the mechanical erosion. One is the hydraulic model experiment with a tracer. Locations and extent of the physical erosion are predicted through analyzing the range and the depth of the tracer color [4]. The other is a numerical method of Computational Fluid Dynamics (CFD) to analyze fluid properties, such as velocity, temperature, pressure drop, viscosity and thermal stress. The hydraulic model experiment has inherent defects, such as high-cost, high-labor and limit-specific results of experiments. Therefore, many scientists choose the numerical method to investigate their work.

Luo et al. [5] applied Ansys commercial software (Fluent) to analyze velocity distribution of molten slag and hot metal in a main trough. The results show that the fluid's velocities in the center of the trough are faster than ones near the wall and depend on the shape of the trough. Dash et al. [6] studied the fluid and turbulent kinetic energy in a main trough by the numerical analysis and investigated the effect of the slope of the main trough on the velocity distribution. Luomala et al. [7] used CFD and a 1/4 scaled-down hydraulic model with a laser Doppler velocimeter to study the properties of fluid in the main trough and the effect of the dam height on velocity distribution. Duan et al. [3] calculated the temperature distribution of a main trough using a three-dimensional (3D) model considering natural convection and forced convection and proposed that a new main trough be designed based on the gradient arrangement of the bricks. Wang et al. [8] combined the turbulent model and the volume of fraction (VOF) to develop a 3D fluid model of a main trough and studied the effects of the tap hole stream velocity and the trough geometry on the fluid flow. Chang et al. [9] used a momentum conservation equation and VOF to analyze a main trough flow velocity and wall shear stress, and proposed a method to reduce the refractory wear of the blast furnace. The above literature only concentrates on the flow properties (velocity, pressure, viscosity and so on), temperature distribution in the trough and studies the influence of the trough structure on the fluid. However, the effect of the hot metal trajectory leaving the tap hole on the velocity and the temperature of a trough and the refractory erosion during tapping are not reported. Therefore, this investigation will focus on the effect of the hot metal trajectory.

In this paper, OpenFoam is used to solve the transient Navier–Stocks equations including the mass, momentum and energy conservation equations. In Section 2, the solved issue will be addressed. Then, a mathematical model, boundary conditions and solution of the mathematical model are presented in detail. In Section 2, calculation results are presented and discussed. For example, the velocity, the temperature and the wall shear stress of the main trough are analyzed at different tapping moments. Furthermore, the shear stress under different tap hole angles is analyzed and temperature in the refractory is studied by conjugate heat transfer between the refractory and the fluid. In the last section, the conclusions from the work are summarized.

## 2. Problem Formulation

### 2.1. Physical Model

According to the shape of a blast furnace trough from a steel plant in China, a physical model is established in Figure 1. During tapping, molten slag and hot metal are regarded as a mixed continuous and incompressible fluid flowing out from the tap hole, and then fall down into the main trough. The mixture fluid keeps a constant level in the main trough and around 300 mm from the upper surface in the calculation. It is separated by the skimmer, and then hot metal flows into a torpedo. Falling position of the mixture fluid trajectory (FPMFT) in the trough defines the inlet of the model. It moves from 4 m away from the origin of the coordinates in the beginning to the tap hole direction in the tapping process. According to References [5,10], Table 1 lists the physical properties of the mixture fluid and the refractory in the study. Chemical reaction between the refractory and the mixture fluid is

neglected in simulations. The diameter and the angle of the tap hole is 60 mm and 10 degrees in the simulations, respectively.

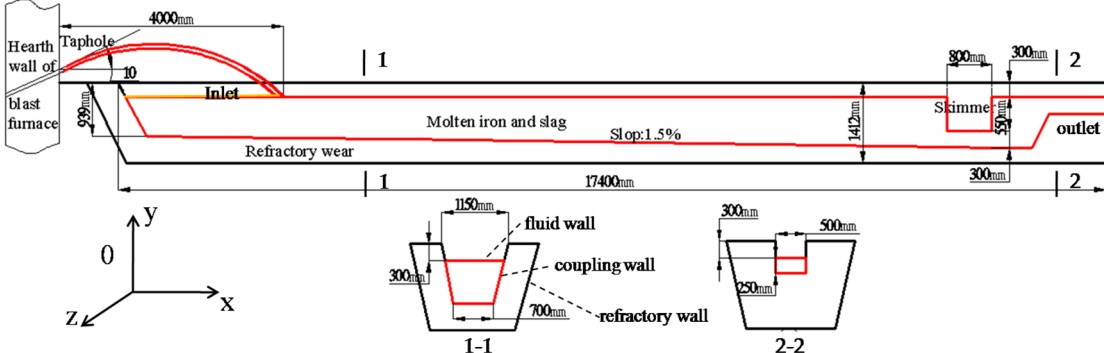

**Figure 1.** Schematic diagram of the main trough in front view: 1-1 cross-section view at the main trough; 2-2 cross-section view at the outlet.

**Table 1.** Physical properties of the mixture and the refractory.

| Property | Value |
|---|---|
| Temperature of the inlet (K) | 1773 |
| Temperature of the main trough (K) | 1583 |
| Temperature of the refractory (K) | 1273 |
| Thermal conductivity of the fluid ($kg \cdot m \cdot s^{-3} \cdot K^{-1}$) | 33 |
| Thermal conductivity of the refractory ($kg \cdot m \cdot s^{-3} \cdot K^{-1}$) | 0.16 |
| Density of iron ($kg \cdot m^{-3}$) | 6900 |
| Density of slag ($kg \cdot m^{-3}$) | 2600 |
| Viscosity of iron ($kg \cdot m^{-1} \cdot s^{-1}$) | 0.0045 |
| Viscosity of slag ($kg \cdot m^{-1} \cdot s^{-1}$) | 0.25 |

Where temperature and viscosity are given in constant values for calculation speed, and the effect of them on the simulation will be focused on later.

*2.2. Mathematical Model*

2.2.1. Mathematical Model of Molten Slag and Hot Metal

The governing equations of the mixture fluid include a mass conservation equation, a momentum equation based on Reynolds-averaged one, and an energy conservation equation. The fluid in the study was an incompressible Newtonian fluid and its volume expansion ratio $\frac{\partial u_i}{\partial x_i}$ is zero. In order to maintain the conservation of the mixture, the mass conservation equation must be met [11], as:

$$\frac{\partial u_i}{\partial x_i} = 0, \tag{1}$$

where, $x_i$ and $u_i$ express the coordinates of space points (m) and the velocity component at point $x_i$ of the time t coordinate ($m \cdot s^{-1}$), respectively. $x_1$, $x_2$ and $x_3$ define the three directions of x, y and z, respectively.

The viscous stress tensor P and the deformation rate tensor S of Newtonian fluid have a linear and isotropic function relationship [12]. The Newtonian fluid constitutive equation is substituted into the dynamic equation to obtain the momentum conservation equation of the incompressible Newtonian fluid [11], as:

$$\frac{\partial u_i}{\partial t} + u_j \frac{\partial u_i}{\partial x_j} = -\frac{1}{\rho} \frac{\partial p}{\partial x_i} + v \frac{\partial}{\partial x_j} \left( \frac{\partial u_i}{\partial x_j} \right), \tag{2}$$

where, $p$, $\rho$, $\mu$ and $\nu$ express pressure of the fluid (kg·m$^{-1}$·s$^{-2}$), viscosity of the fluid (kg·m$^{-3}$), kinetic viscosity (kg·m$^{-1}$·s$^{-1}$) and kinematic viscosity (m$^2$·s$^{-1}$), respectively.

In this study, the standard $k$-$\varepsilon$ turbulence model is used in the simulation. The momentum conservation equation is a time average one to obtain the Reynolds-averaged N-S equation [13]:

$$\frac{\partial \langle u_i \rangle}{\partial t} + \langle u_j \rangle \frac{\partial \langle u_i \rangle}{\partial x_j} + \frac{\partial \langle u_i' u_j' \rangle}{\partial x_j} = -\frac{1}{\rho}\frac{\partial \langle p \rangle}{\partial x_i} + \nu \frac{\partial}{\partial x_j}\left(\frac{\partial \langle u_i \rangle}{\partial x_j}\right), \tag{3}$$

where, $u_i'$ and $u_j'$ define pulse values of the velocity (m·s$^{-1}$), respectively. $\langle u_i \rangle$ is time average, and $u_i = \langle u_i \rangle + u_i'$.

Reynolds stress tensor term, $-\langle u_i' u_j' \rangle$, is added into Equation (3). This makes the equations disable to close and introduces a turbulence model. According to the Boussinesq hypothesis, the expression of Reynolds [13] stress is:

$$-\langle u_i' u_j' \rangle = \nu_t\left[\frac{\partial}{\partial x_j}\langle u_i \rangle + \frac{\partial}{\partial x_i}u_j\right] - \frac{2}{3}\delta_{ij}k, \tag{4}$$

where, $\delta_{ij}$ is Kronecker delta and $\delta_{ij} = \begin{cases} 1, & i = j \\ 0, & i \neq j \end{cases}$. $k$ is turbulent energy (m$^2$·s$^{-2}$).

Due to high velocity at the inlet, the mixture fluid in the main trough has a high Reynolds number. The standard $k-\varepsilon$ turbulence model has a few empirical constants for this condition. In the standard $k-\varepsilon$ model, the turbulent energy k and the turbulent dissipation rate $\varepsilon$ are associated with the turbulence $\nu_t$, the formula [13] is as follows:

$$\nu_t = C_\mu \frac{k^2}{\varepsilon}, \tag{5}$$

where, $C_\mu$ expresses the empirical constant and a value of 0.09 is used in the simulation.

$k$ and $\varepsilon$ are solved in an incompressible fluid using the following two equations [13]:

$$\frac{\partial(\rho k)}{\partial t} + \langle u_i \rangle\frac{\partial(\rho k)}{\partial x_i} = \frac{\partial}{\partial x_i}\left[\left(\mu + \frac{\nu_t}{\sigma_k}\right)\frac{\partial k}{\partial x_j}\right] + G_k - \rho\varepsilon, \tag{6}$$

$$\frac{\partial(\rho\varepsilon)}{\partial t} + \langle u_i \rangle\frac{\partial(\rho\varepsilon)}{\partial x_i} = \frac{\partial}{\partial x_i}\left[\left(\mu + \frac{\nu_t}{\sigma_\varepsilon}\right)\frac{\partial\varepsilon}{\partial x_i}\right] + \frac{C_{1\varepsilon}\varepsilon}{k}G_k - C_{2\varepsilon}\rho\frac{\varepsilon^2}{k}, \tag{7}$$

where, $C_{1\varepsilon}$, $C_{2\varepsilon}$, $\sigma_k$ and $\sigma_\varepsilon$ express 1.44, 1.92, 1.0 and 1.3, respectively. $G_k$ defines the increase in turbulent kinetic energy caused by the average velocity gradient and is calculated as follows:

$$G_k = \nu_t\left(\frac{\partial \langle u_i \rangle}{\partial u_j} + \frac{\partial \langle u_j \rangle}{\partial u_i}\right)\frac{\partial \langle u_i \rangle}{\partial x_j}, \tag{8}$$

The above eight equations jointly solve the velocity and the pressure of the mixture fluid region and the energy conservation equation is expressed [11] by:

$$\frac{\partial T}{\partial t} + u_i\left(\frac{\partial T}{\partial x_i}\right) = \frac{\partial}{\partial x_i}\frac{\lambda}{\rho C_p}\frac{\partial T}{\partial x_i}, \tag{9}$$

where, $\lambda$ and $C_p$ express the fluid heat transfer coefficient (W·m$^{-1}$·K$^{-1}$) and the specific heat capacity of fluid (J·m$^{-1}$·s$^{-1}$), respectively.

The energy conservation equation is also a time average one. Equation (9) is added to the Reynolds heat conduction term ($\langle u_i' T' \rangle$) after time-average, and it becomes:

$$\frac{D\langle u_i' T' \rangle}{Dt} = \frac{\partial}{\partial x_j}\left[C_T\frac{k^2}{\varepsilon}\frac{\partial \langle u_i' T' \rangle}{\partial x_j} + a\frac{\partial \langle u_i' T' \rangle}{\partial x_j}\right] - \left(\langle u_i' u_j' \rangle\frac{\partial \langle T \rangle}{\partial x_j} + \langle u_j' T' \rangle\frac{\partial \langle u_i \rangle}{\partial x_j}\right) - C_{T1}\frac{\varepsilon}{k}\langle u_i' T' \rangle - C_{T2}\frac{\partial \langle u_i \rangle}{\partial x_j}\langle u_j' T' \rangle, \tag{10}$$

where, $C_T$, $C_{T1}$ and $C_{T2}$ are empirical coefficients. The values of them are 0.07, 3.2 and 0.5 in the simulation, respectively.

### 2.2.2. Mathematical Model of Refractory

Heat transfer between the mixture fluid and the refractory is coupled to each other. For the refractory heat transfer calculations, only the Fourier equation [14] is solved:

$$\frac{\partial \langle T \rangle}{\partial t} = \frac{\partial}{\partial x_i} \frac{\lambda}{\rho C_p} \frac{\partial \langle T \rangle}{\partial x_i}, \tag{11}$$

where, $\lambda$ defines the solid heat transfer coefficient (W m$^{-1}$·K$^{-1}$). $C_p$ expresses the specific heat of the refractory (Al$_2$O$_3$–SiC–C) and is 0.628 kJ·kg$^{-1}$·K$^{-1}$.

### 2.3. Boundary Conditions (cf. Figure 2)

(1)  Inlet boundary conditions. Due to the decrease of the pressure in the furnace during tapping, mass flow of the mixture flow from the tap hole decreases and FPMFT moves to the tap hole direction. Therefore, boundary condition at the inlet is velocity type and it can change in the direction and in the magnitude at the same time. A parabolic Equation (12) is used to define the velocity. Its maximum magnitude is 6.635 m·s$^{-1}$ and is estimated from the FPMFT at the beginning of the tapping. Thermal and pressure boundary conditions at the inlet are constants of 1773 K and zero gradient, respectively.

$$y = x \cdot \tan \alpha - \frac{g}{2u_0^2 \cos^2 \alpha} \cdot x^2, \tag{12}$$

where, $\alpha$ is the inclined angle of the tap hole (°), $u_0$ defines the velocity of the mixture fluid stream (m·s$^{-1}$) and changes with the time, x and y are the coordinates of FPMFT (m, m) and g expresses the acceleration of gravity (m·s$^{-2}$).

(2)  Outlet boundary conditions. The outlet of the main trough defines pressure type and is expressed as zero. Thermal and velocity boundary conditions are constants of 1583 K and zero gradient, respectively.

(3)  Wall boundary conditions. Boundary condition for free surface of the main trough is no slip. The temperature and pressure are constants of 1583 K and zero gradient at the walls, respectively.

(4)  Interaction wall boundary conditions (see yellow surface in Figure 2) for the mixture fluid and the refractory. Temperature boundary condition is a conjugate heat transfer. Velocity and pressure are constants of zero and zero gradient at the interaction wall, respectively.

(5)  Refractory wall boundary conditions. Since the refractory only needs to solve the Fourier's equation, there is only a temperature boundary condition with a constant of 1273 K.

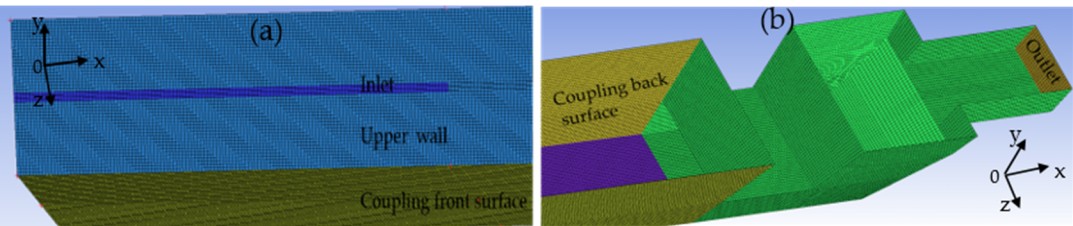

**Figure 2.** Computational grid: (**a**) the grids at the inlet and the upper wall, (**b**) the grids at the outlet.

### 2.4. Numerical Procedure

(1)  Pre-process. A 3D modeling software is employed to draw a geometry. Hexahedron structure meshes of the geometry are created by the Integrated Computer Engineering and Manufacturing

(ICEM) and are shown in Figure 2. There are 6,433,548 grids, including 1,477,213 for the mixture fluid and 4,956,335 for the refractory.

(2) Solution. The mixture fluid in the main trough is solved by Equations (1), (3) and (9), and the solid in the refractory is solved by Equation (11). For transient simulation, these equations are discretized in time and a time step of 0.001 s is used. In every time step, the simulation is solved by the semi-implicit method for pressure-linked equations (SIMPLE) algorithm as a steady state. Then, the pressure implicit with splitting of operator (PISO) is employed to calculate the transient discretization until the last time step.

(3) Post processing. Paraview and Tecplot software is used to visualize the simulations. Locations of 5 faces and 6 lines in the main trough are shown in Figure 3a, b to analyze the results. A center plane (red) is the central cross-section of the main trough. The distance between the center plane and plane 1 (blue) and one between plane 1 and 2 (orange) are both 0.2 m. Plane 3 (green) defines the back surface of the mixture fluid in the main trough. Plane 4 (deep blue) is perpendicular to other planes with a horizontal distance of 3 m from the origin of the coordinates and also includes some parts in the refractory. Except for line 2, other lines are located on the center plane. Line 1 expresses the intersection of the central plane and the bottom surface of the main trough. The distance between line 1 and 3, one between line 3 and 4, one between line 4 and 5 and one between line 5 and 6 are 0.02, 0.2, 0.2 and 0.2 m, respectively. The intersection of the front surface (cf. Figure 2a) and the bottom surface of the main trough defines line 2.

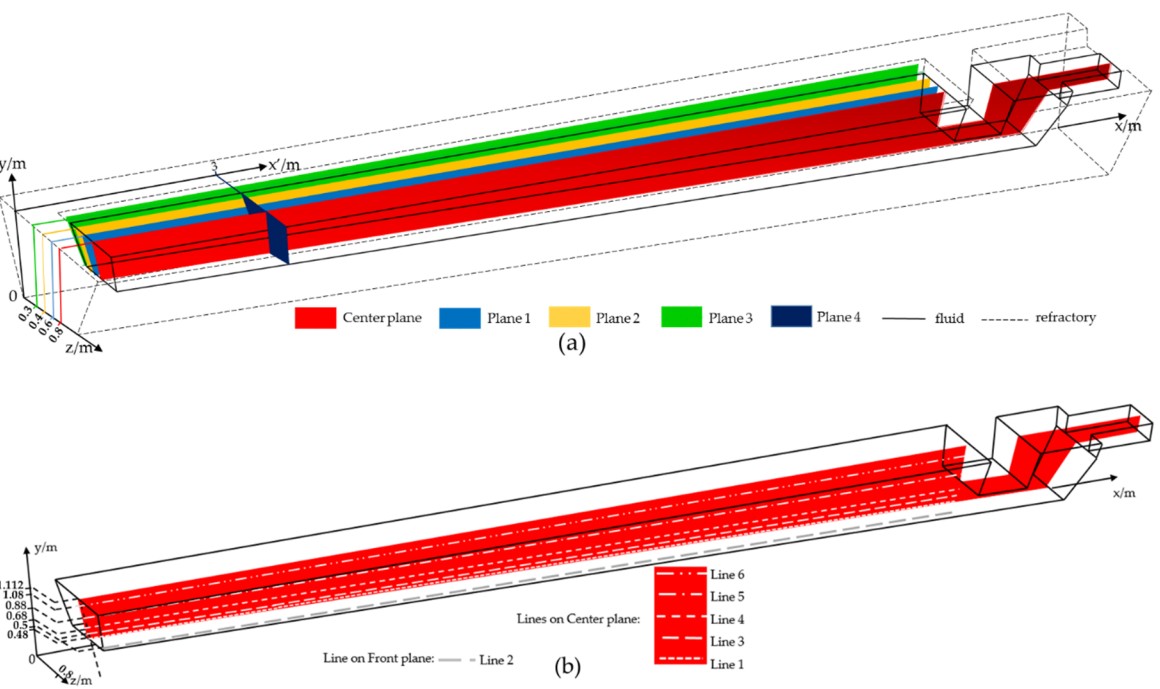

**Figure 3.** Locations of 5 faces and 6 lines in the main trough for post-processing analysis: (**a**) planes in the full main trough, (**b**) lines in the mixture fluid of the main trough.

## 3. Results and Discussion

### 3.1. Residual Errors in Simulation

The residuals include internal and external, two parts in the simulation. When transient state calculations happen in OpenFoam, the number of iterations in each time step is controlled by the internal residuals. The external residual is the difference between the calculated results at adjacent time steps. When the convergence criterion is reached, the calculation moves to the next time simulation. The smaller the residual errors are, the better convergence the calculations have. Figure 4 shows

residual error values of pressure, velocity and temperature at the initial moment of calculation. It can be observed that with the increase of the calculation time, the residuals errors gradually decrease. All of them meet the convergence criteria and the calculation results are reliable.

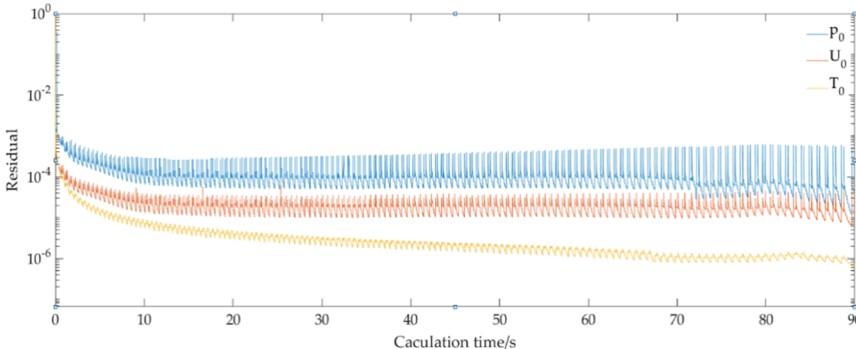

**Figure 4.** Monitoring residual error curves of pressure, velocity and temperature in the simulation.

### 3.2. Velocity Distribution of the Mixture Fluid in the Main Trough

Tapping lasts around 90 min in practice and it is not necessary to simulate the full process. Because, except for the beginning of 1 to 2 min and the end of 1 to 2 min, molten slag and hot metal keep a constant mass flow and flow out from the tap holes. Simulation should include the following three steps: the start moment, constant state and the end. Therefore, the total calculation is 90 s and includes the three stages. Furthermore, time step is 0.001 s and the courant number is smaller than 1.

The calculation results at 5, 30, 55 and 80 s are used to analyze the velocity distribution of the mixture fluid at the initial, early intermedia, late intermedia and the end of the tapping. Figure 5 shows the velocity vector of the center plane (cf. Figure 3a) along the main trough. The FPMFT is 2.7 m from the origin of the coordinates in the main trough and downstream of the FPMFT is strongly influenced by the tap hole flow at 5 s. Therefore, a counter clockwise turbulence is observed near 4.5 m, but the impact of the turbulence on the upstream of the FPMFT is quite weak.

The FPMFT moves to the tap hole direction during tapping. The turbulence of FPMFT's downstream is fully developed and its influence range is obviously increased at 30 s. It decreases a lot at 50 s and disappears at 80 s. Comparing four figures in Figure 5, the angle of the mixture fluid at the moving inlet (cf. Equation (12)) changes hugely, and the mixture fluid flow hits the bottom of the main trough. Therefore, more hot metal and molten slag mechanically erode the bottom wall of the refractory.

Figure 5 shows that the erosion near the FPMFT is more serious and the velocity distribution at the FPMFT of 3 m is studied. Figure 6 shows the velocity vector at 3 m from the origin of the coordinates in the main trough at 5, 30, 55 and 80 s. With the increase of the time, the velocity on plane 4 is also significantly reduced. The velocity at the center of plane 4 is large in the period of 2 to 2.6 m/s at 5 s. At 30 s, obvious turbulence is observed to form in the main trough and velocity at the bottom of the main trough is the largest (around 0.6 m/s). Therefore, physical erosion at the bottom of the main trough is more possible and serious. At 55 s, the mixture fluid velocity in the lower side walls of the main trough is larger than other locations. At 80 s, two "donuts" flows appear on the cross-section, but velocity magnitude of the flow is quite small (maximum 0.05 m/s). Therefore, the mixture fluid flow has little effect on the main trough.

Comparing maximum velocities (red arrows) in Figures 5 and 6, velocities of the mixture fluid from the inlet become smaller and smaller during the tapping, which means that mechanical erosion mainly happens in the beginning of the tapping.

In summary, the mixture fluid flow of the main trough is significantly affected by the fluid from the inlet, and a strong turbulence is formed at the downstream of FPMFT and the turbulent area also

expands toward the skimmer. The turbulent area near the initial moment of FPHMT exists for a long time. The velocity at the bottom and the lower side walls of the main trough is bigger than others.

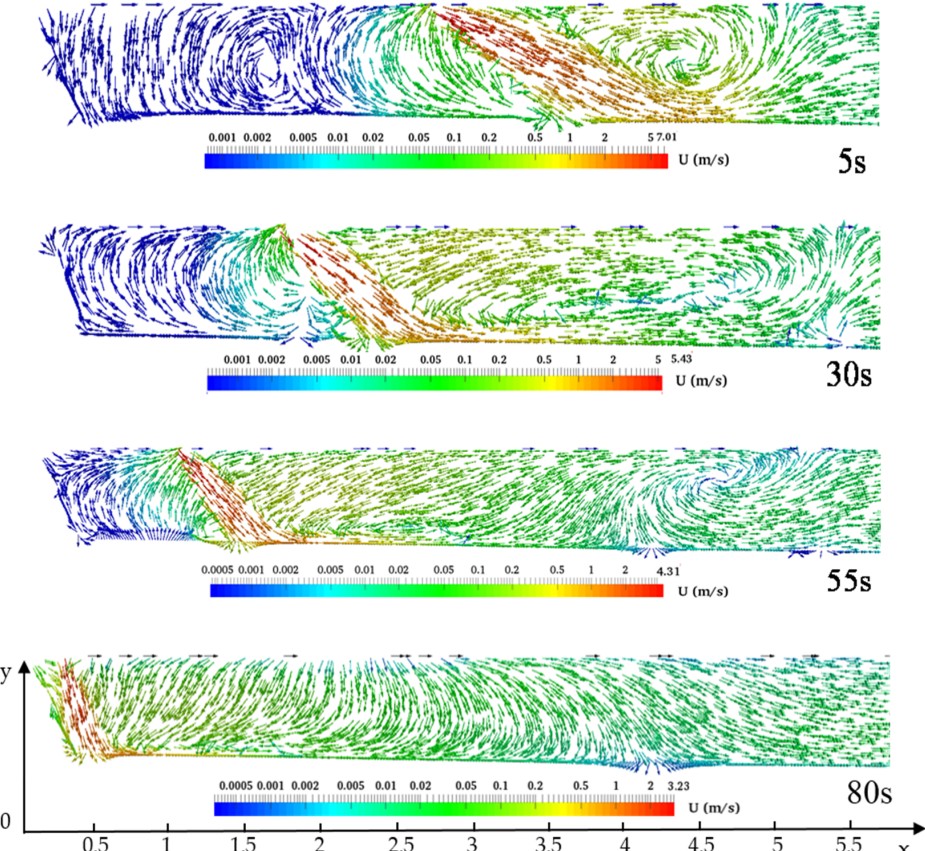

**Figure 5.** Velocity distribution on the center plane (cf. Figure 3a) at 5, 30, 55 and 80 s.

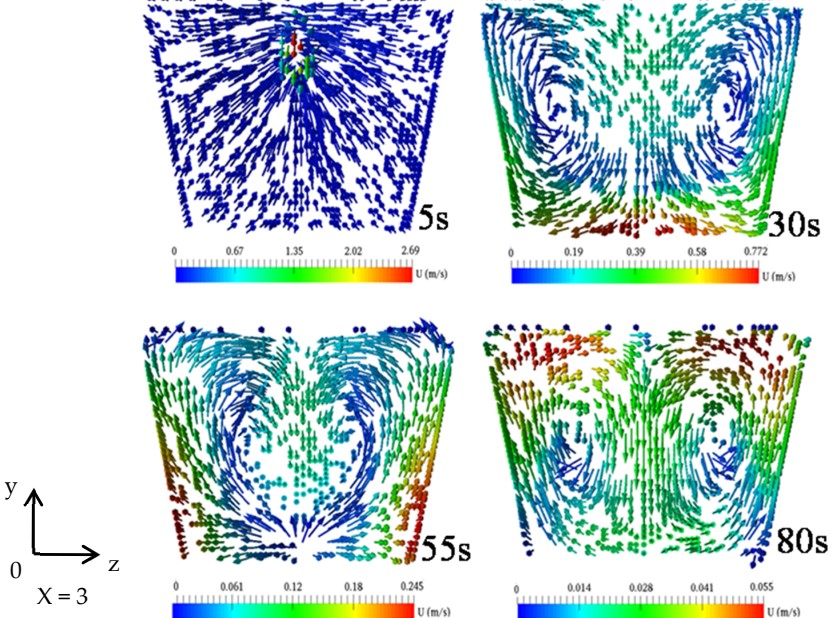

**Figure 6.** Velocity distribution of the mixture fluid on plane 4 (cf. Figure 3a) at 5, 30, 55 and 80 s.

### 3.3. Wall Shear Stress

In order to quantitatively study and describe mechanical erosion, wall shear stress is chosen and expressed by:

$$\tau = -\mu \frac{\overrightarrow{u}}{\overrightarrow{n}}, \tag{13}$$

where, $\overrightarrow{n}$ and $\mu$ indicate the normal vector (m) and the mixed viscosity of molten iron and slag (kg·m$^{-1}$·s$^{-1}$), respectively. $\frac{\overrightarrow{u}}{\overrightarrow{n}}$ is change rate of velocity perpendicular to the direction of the fluid movement.

Figure 7 shows wall shear stress distribution of the refractory on line 1 and line 2 at 5 s. Wall shear stress reaches the maximum at 4 m, and its value at the bottom (line 1) of the main trough is significantly larger than the side wall (line 2). The FPMFT at 2.7 m and the velocity at 4 m is strongly influenced by the mixture fluid flow from the inlet at the moment. Due to effect of the skimmer, the wall shear stress increases a little at 15 m but there is no difference between the side wall and the bottom wall.

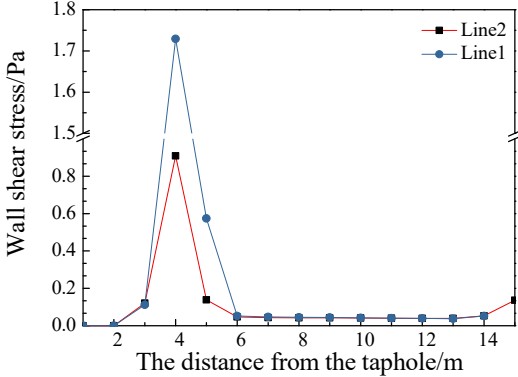

**Figure 7.** Wall shear stress on the line 1 (blue curve, cf. Figure 3b) and line 2 (red curve, cf. Figure 3b).

### 3.4. Temperature Distribution of the Main Trough

Figure 8 shows the temperature distribution of the central plane (cf. Figure 3a) at 5, 30, 55 and 80 s. The temperature distribution matches the velocity distribution in the previous section. The downstream of the FPMFT has a significant increase of the temperature due to the mixture fluid from the inlet with high energy, while the temperature of the upstream mixture changes little.

Figure 9 shows the temperature distribution of the mixture fluid on plane 4 (cf. Figure 3a) at 5, 30 and 55 s. At 5 s, the mixture fluid flows from the tap hole and a high temperature is concentrated on the center of the cross-section. At 30 s, the bottom of the main trough forms a higher temperature region than others. The temperature at the others is around 1605 K and is much higher than that at 5 s. At 55 s, the falling position of the mixture fluid is located at 1.2 m and the high-temperature zone is located at the side and the bottom walls of the main trough.

Figure 10 show temperature changes of four lines during tapping ((a) line 3, (b) line 4, (c) line 5 and (d) line 6, cf. Figure 3b). Temperature varies greatly on the main trough direction, especially when the mixture fluid is close to the FPMFT. Due to movement of the FPMFT, temperature varies greatly before 3 m. There is a constant temperature zone of 1610 k from 3 to 6 m. The largest fluctuation happens near the bottom of the mixture fluid (Figure 10a), which means the bottom refractory suffers the highest frequent changes of thermal stress and is highly probable to erode.

Figure 11 shows the temperature distribution of the mixture fluid from the central plane to plane 3 of the main trough (cf. Figure 3a). Temperature near FPMFT is obviously higher than others, and temperature at the downstream of FPMFT is apparently higher than the initial boundary set 1573 K due to the conduction and the convection of heat transfer. Furthermore, the temperature gradually decreases from the central plane trough (z = 0.8) to the plane 3 (z = 0.3). Due to a low thermal

conductivity of the refractory and the low initial temperature, the temperature gradient in the refractory is really small. However, the temperature at the interaction boundary wall of the refractory and the mixture fluid is exactly the same as each other.

In summary, the temperature of the mixture fluid increases significantly during tapping. From the center of the mixture fluid to the refractory, the temperature gradually decreases, but the temperature distribution is consistent with the velocity field. In the vertical direction, temperature increases with the increase of the height. In the horizontal direction, the temperature of the mixture fluid near FPMFT changes greatly.

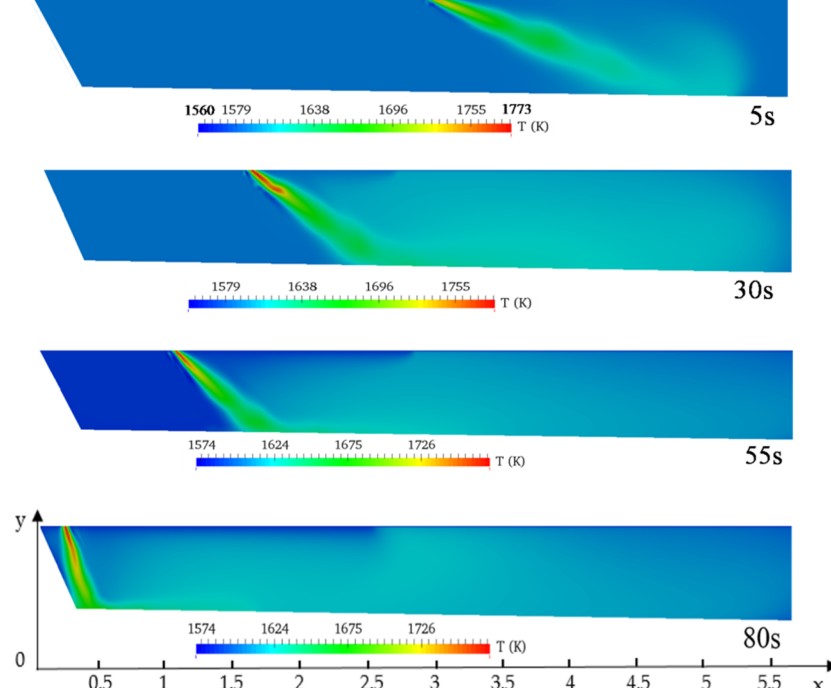

**Figure 8.** Temperature distribution of the center plane (cf. Figure 3a) at 5, 30, 55 and 80 s.

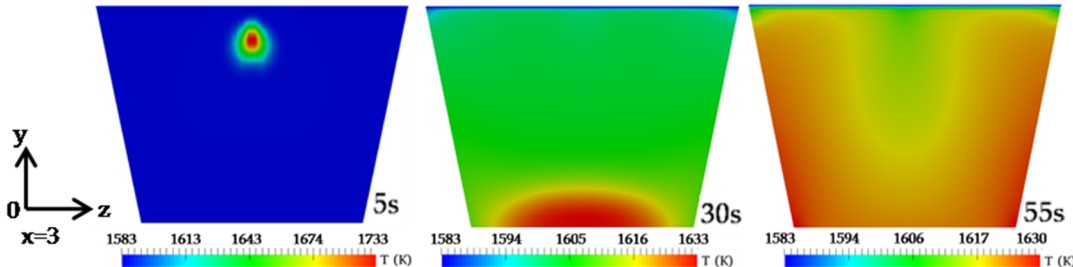

**Figure 9.** Temperature distribution of the mixture fluid on plane 4 (cf. Figure 3a) at 5, 30 and 55 s.

### 3.5. Influence of Tap hole Angles on the Flow of the Main Trough

Intermittent tapping of a blast furnace must be punched before the tapping and be plugged after the tapping. The angle of the tap hole can change during tapping. Therefore, it is important to understand the influence of the angle of the tap hole to the temperature, velocity and shear stress of the mixture flow in the main trough. Since velocity and temperature distributions are quite similar in Figures 5–11, they are not analyzed any more.

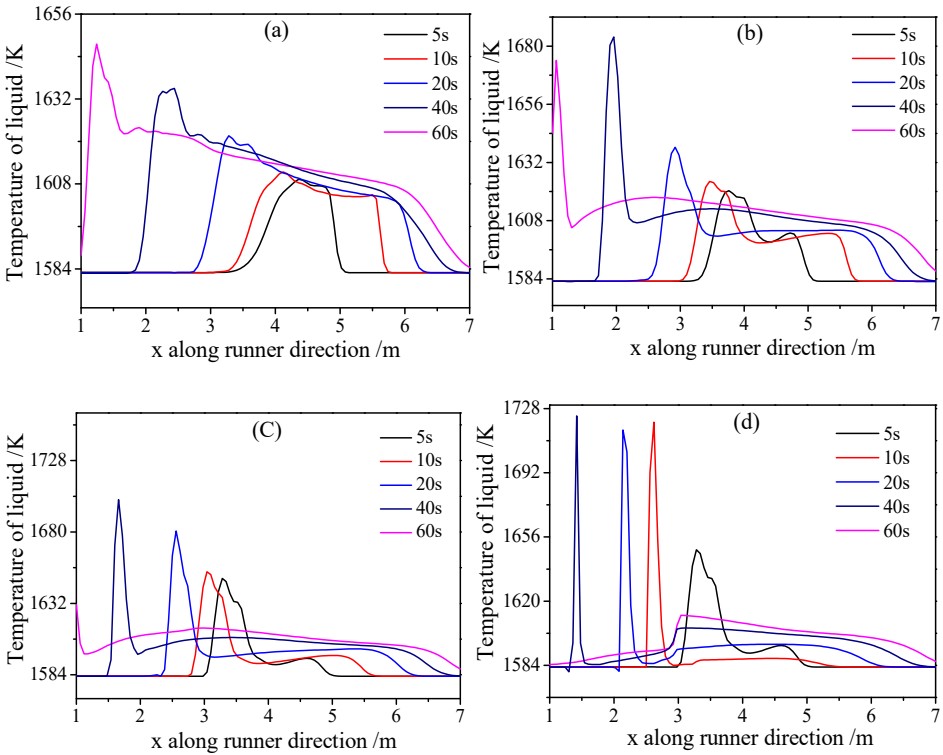

**Figure 10.** Temperature fluctuations of four lines during tapping ((**a**) line 3, (**b**) line 4, (**c**) line 5 and (**d**) line 6, cf. Figure 3b).

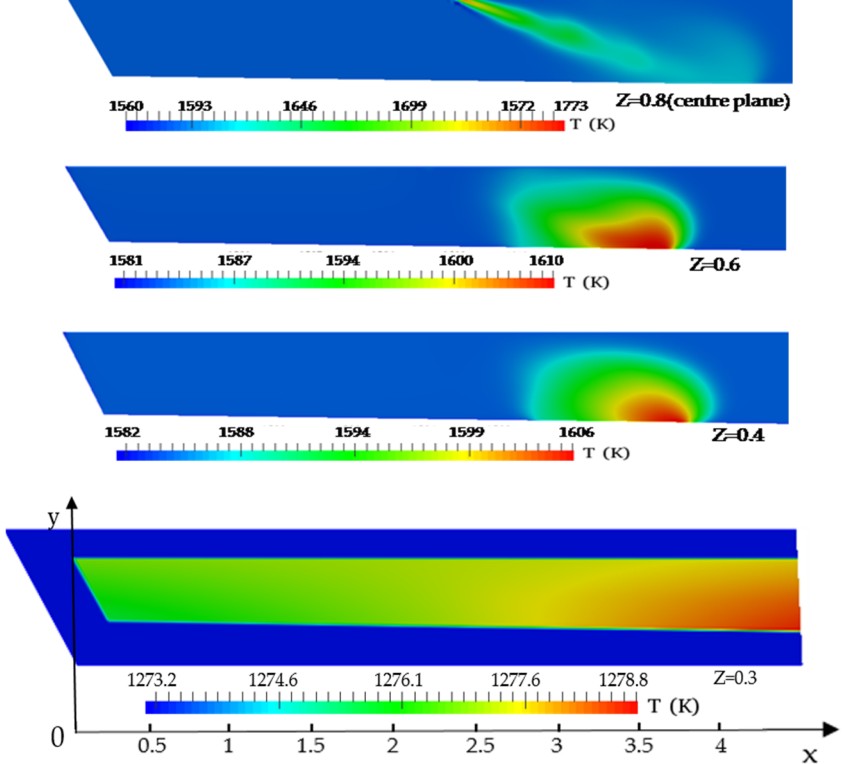

**Figure 11.** Temperature distributions from the central plane to plane 3 (cf. Figure 3a).

Generally speaking, the angle of the tap hole changes between 7 to 12°. Therefore, two values in the interval were selected to highlight the effect of the angle. As shown in Figure 12, with the increase of the tap hole angle, the wall shear stress increases from 0.73 to 0.87 because molten slag and hot

metal gets a higher velocity when they fall into the main trough from a bigger angle of the tap hole. The little increase of wall shear stress happens at 15 m due to the fact that the fluid velocity becomes bigger at the skimmer. When the tap hole angle is 7°, mechanical erosion of the trough has the smallest value. The maximum of wall shear stress is reduced by 16% and the campaign of the main trough is estimated to expand over 5 days, comparing with the tap hole angle of 10° and 7°.

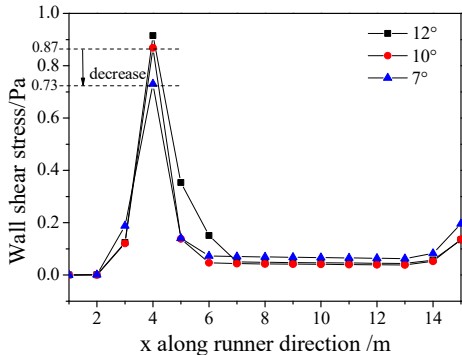

**Figure 12.** Wall shear stress distribution at different angles of the tap hole on line 2 (cf. Figure 3b).

Figure 13 shows temperature changes of line 3 during tapping with different tap hole angles, (a) 7°, (b) 10° and (c) 12°. With the increase of the angle of the tap hole, change of temperature during tapping is little. But as the location of FPMFT moves toward the direction of the tap hole, the maximum temperature of it increases. During tapping, reasonable adjustment of the tap hole angle can reduce the extent of mechanical erosion in some areas, which is helpful to expand the sieve campaign of the main trough.

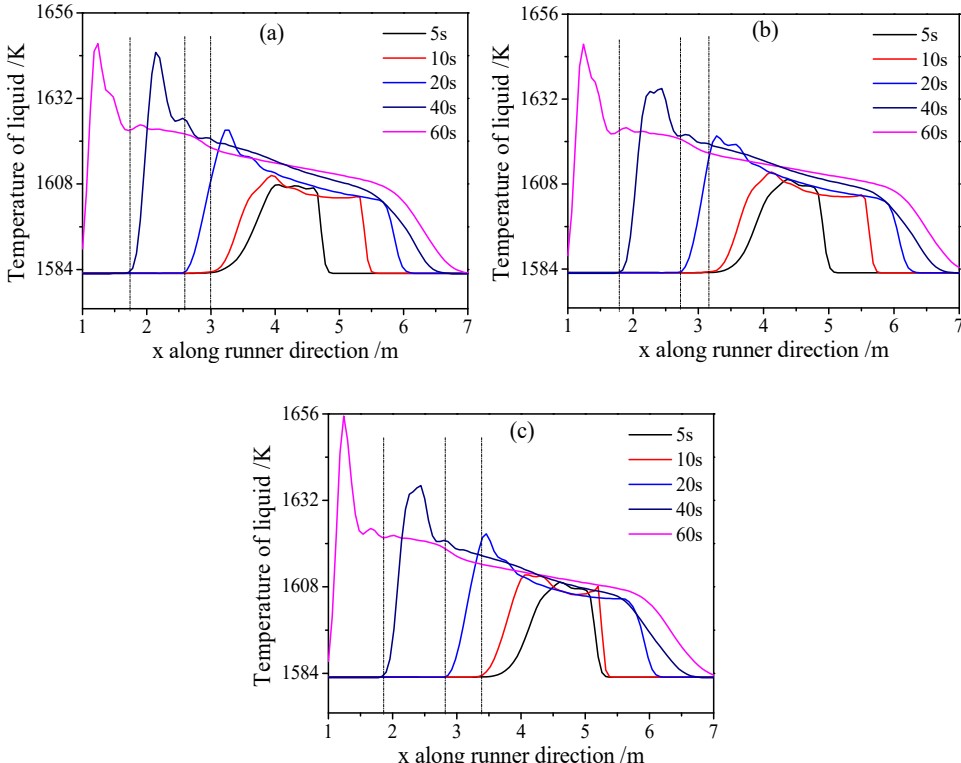

**Figure 13.** Temperature fluctuations on line 3 (cf. Figure 3b) during tapping with different tap hole angles, (**a**) 7°, (**b**) 10° and (**c**) 12°.

## 4. Conclusions

In this study, OpenFoam was employed to analyze the velocity, the temperature, the wall shear stress in the main trough of a blast furnace and the influence of different tap hole angles on the main trough. Some conclusions are highlighted as follows.

(1)  Velocity of the mixture fluid at the center of the main trough is generally larger than that at the wall. Velocity of the FPMFT at the downstream is larger than that at the upstream. However, due to strong turbulence at the downstream, the lower front wall also has larger velocity.

(2)  Maximum of shear stress occurs at the downstream of the FPMFT, and the shear stress on the bottom wall is bigger than that on the front wall.

(3)  Due to velocity increase of the FPMFT, the shear stress on the wall of the main trough increases with the increase of the tap hole angle. When the tap hole angle is 7°, mechanical erosion of the trough has the smallest value and the campaign of the trough expands over 5 days.

**Author Contributions:** Y.G. is responsible for writing the article and simulation calculation of blast furnace main trough. M.L. is responsible for the guidance of code writing in simulation. H.W. is responsible for the language and format of the article. D.L. and X.W. provide blast furnace main trough data involved in the simulation process. Y.Y. guide the overall direction of the article, provide simulation technical guidance and simulation equipment. All authors have read and agreed to the published version of the manuscript.

**Funding:** We gratefully acknowledge financial support from the Program for Professor of Special Appointment (Eastern Scholar) at Shanghai Institutions of Higher Learning (No.TP2015039), National Natural Science Foundation of China (No.51974182), National 111 project, Grant/Award No. 17002 and Laiwu Steel. The simulations and analyses were completed using OpenFoam open source.

**Conflicts of Interest:** The authors declare no conflict of interest.

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
