# Peer review of "Numerical Analysis on Velocity and Temperature of the Fluid in a Blast Furnace Main Trough"

_processes, doi:10.3390/pr8020249_

Round 1

Reviewer 1 Report

Dear Authors,

Your paper “Numerical Analysis on the flow and temperature in Blast Furnace Main Trough” discusses about the CFD simulation of the iron and slag tapping from blast furnace with respect the erosion and temperature stress of the main refractory tough.

The introduction gives a good literature review, mainly focusing on the difference between Authors and others simulations performed on the same topic. However the importance to fully understand what happen during tapping is not well defined, especially in terms of productivity, residual live of the though and working campaign of the blast furnace. Please, add this information to your introduction in order to better highglight the significance and the novelty of your work.

Model design is accurate and well explained, even some aspect must be clarified, as reported below.

Conclusions are supported by the results.

Thus, my evaluation is major revision

Please find in the following some comments/remarks/corrections, to apply to your paper before resubmission

P.2, line 59-60. Please, check the spacing between words. I highlighted in bold were changes must be made. “However, effect of hot metal trajectory leaving tapholeon the velocity and temperature of the fluid during tapping in the runner and the refractory arenot reported”. Please check also in the whole text because I found several sections where this problem is.

Table 1. Why the physical properties of steel and slag were assumed as constants? During the flowing, a temperature decreases occurs and this modified density and viscosity. Do the Authors think that these variations can affect their simulation?

Equation 10. Which kind of refractory was supposed to be in the main through? Which value of Cp was used? A constant or a temperature-depending function?

Equation 11. Please check the format of alfa

Figure 6: sorry but I cannot identify the two lines referred in the figure caption in the Figure 2. Please check and correct accordingly. Please check also in the other figures referring to Figure 2.

Please add a space between values and unit (i.e. 5 s; 5 m, 1200 K)

Best Regards

Author Response

Dear editor,

   Thanks you for informing us the positive feedbacks; as well as the reviewers for quickly processing reviews of our manuscript. We feel very thanks for reviewers professional review work on our article. As they concerned, we have made extensive corrections to our previous draft, the detailed corrections are listed below.

Reviewer #1:

The introduction gives a good literature review, mainly focusing on the difference between Authors and others simulations performed on the same topic. However the importance to fully understand what happen during tapping is not well defined, especially in terms of productivity, residual live of the though and working campaign of the blast furnace. Please, add this information to your introduction in order to better highglight the significance and the novelty of your work.

Response:  Thanks for your suggestion. We added the following to introduce the tapping of the blast furnace. " In tapping of 3000 m3 blast furnace, 4 to 7 tons per minute of molten slag and hot metal with the temperature = 1773 K flow into the main trough from the taphole and tapping time is 70 to 120 minutes and tapping number per day is around 15. Then, under effect of gravity force, the molten slag moves to the skimmer on top of hot metal and is separated into slag trough by the skimmer. Due to the harsh working environment of main trough, its average service life is around 35 days. During the life, the trough of 3000 m3 blas furnace needs to be small repaired at least 5 times and ramming material needs 3 tons per time (ramming material price 823 $ per ton). Furthermore main trough has 9 to 10 service life per year and needs 45 tons of casting material (price 857 $ per ton). In short, cost of blast furnace trough is 0.52 million $ per year excluding manpower, time cost and environmental cost. Therefore maintenance cost of main trough is really expensive. Therefore the internal environment of main trough should be known and the erosion mechanism of refractory materials must be understood by operators and managers." We have added this part in lines 25-37 in red.

2, line 59-60. Please, check the spacing between words. I highlighted in bold were changes must be made. “However, effect of hot metal trajectory leaving tapholeon the velocity and temperature of the fluid during tapping in the runner and the refractory arenot reported”. Please check also in the whole text because I found several sections where this problem is.

Response:  We sincerely thank the reviewer for careful reading. We feel so sorry for our carelessness. We have correct the " tapholeon " , "arenot" and "oneand" to " taphole on ",  "are not" and " one and " in the Highlight as well as the whole manuscript in red.

Table 1. Why the physical properties of steel and slag were assumed as constants? During the flowing, a temperature decreases occurs and this modified density and viscosity. Do the Authors think that these variations can affect their simulation?

Response: Thanks, the decrease of the temperature do cause the change of density and viscosity. However, for temperature, change range is around 1573K~1773K and the change of density caused by temperature in this range is not obvious. Therefore, it has little effect on the internal field of the main trough. For viscosity, it has been found that viscosity changes will have a certain effect on the separation of iron and slag, which wasn’t considered in this study. We only studied physical erosion in this paper and consider effect of various parameters on the simulation in the next. Furthermore the change of physical properties will increase the calculation time. Therefore, the parameters as a constant is also helpful to simplify the calculation. We will consider the tempearture effect on molten slag and hot metal properties in another publications. The above lines also add after Table 1 in red.

Equation 10. Which kind of refractory was supposed to be in the main through? Which value of Cp was used? A constant or a temperature-depending function?

Response:  Thanks, the data provided by the steel mill shows that the refractory uses a main trough refractory castable (Al2O3-SiC-C) whose value is set to a constant 0.628 kJ/(kg·â„ƒ). We have added the data in lines 141-142.

Equation 11. Please check the format of alfa.

Response:  Thanks, the format of alfa in Equation 11. has been modified to the correct format Line 151.

Figure 6: sorry but I cannot identify the two lines referred in the figure caption in the Figure 2. Please check and correct accordingly. Please check also in the other figures referring to Figure 2.

Response:  Thanks, we have changed Figure 2 to Figure 3 and highlight the lines and planes used later. Line 1 is the intersection of the center plane and the bottom surface of the main trough. Pass the line 3 to make the parallel surface at the bottom surface of the main trough. The intersection of the parallel surface and the coupling front surface is the line 2. Lines 177-189.

Please add a space between values and unit (i.e. 5 s; 5 m, 1200 K)

Response:  Thanks, we have added spaces between all numbers and units in the article.

Reviewer 2 Report

The issues presented for review are interesting. They present a description of the process of simulating the work of the riverbed and gutter for removing pig iron and slag from the blast furnace. In my opinion, they require comparison with real - practical conditions. I think it will be the subject of another publication.

The Authors did not avoid minor typing errors in the text. Care must be taken in editing the text. Detailed comments in the attached file.

Author Response

Dear editor,

   Thanks you for informing us the positive feedbacks; as well as the reviewers for quickly processing reviews of our manuscript. We feel very thanks for reviewers professional review work on our article. As they concerned, we have made extensive corrections to our previous draft, the detailed corrections are listed below.

Reviewer #2:

verse 127 – oneand – must be one and

verse 213 - ersion – must be erosion

verse 245 – ofmixture – must be of mixture

verse 254 – flutucation – must be fluctuation

verse 269 – boudary – must be boundary

verses 293, 304 – tempeature – must be temperature

verse 281 – Fig. 11 – Wall shear stress/pa – must be Wall shear stress/Pa

Response:  Thanks for your suggestion. We feel sorry for that we didn’t check the whole manuscript carefully our poor writings. We have improved the manuscript and made some changes. All the changes have been marked in red in the revised paper.

Reviewer 3 Report

I believe that the results obtained in the reviewed article should be interesting if the authors can well explain the problem statement and clearly show the obtained results. This should be done with the necessary completeness, so that the reader does not have questions similar to the above. The article needs to be significantly revised. I can’t this article recommend for publication in this form.

Author Response

Dear editor,

   Thanks you for informing us the positive feedbacks; as well as the reviewers for quickly processing reviews of our manuscript. We feel very thanks for reviewers professional review work on our article. As they concerned, we have made extensive corrections to our previous draft, the detailed corrections are listed below.

Reviewer #3:

It is necessary to decipher the abbreviation CDF, ICEM, PIMPLE, PISO in lines 39, 147, 154, 157.

Response:  Thanks for your suggestion. Abbreviations need to be commented when first mentioned in the text. Amend as below: Computational Fluid Dynamics (CFD), Integrated Computer Engineering and Manufacturing (ICEM), PISO and SIMPLE(PIMPLE), the pressure implicit with splitting of operators (PISO) in the text.

In the section entitled "Mathematical model", you must specify that xi are the Cartesian coordinates, and link  them  with  x, y, z ,  as  shown  in    1.  You  must  also  specify what ui , p, μ, ρ, mean.

Response:  Thanks,  is the coordinates of space points,  is the velocity component at point  of the time t coordinate,  ,  and  represent the three directions of x, y and z, respectively. p is pressure of the fluid,  is viscosity of the fluid, is kinetic viscosity.. We also commented the parameters in other formulas for the reader's understanding.  is time average,. is Kronecker delta,  k is turbulent energy. We have added this part in lines 106, 107, 112, 113, 117,121.

It is necessary to put a comma, ending the relations (7) - (10), at some distance from the Otherwise, the denominators in the last terms of these expressions are perceived as , , . In addition, the denominator in the last terms in expressions (9) and (10) must not be , but and the second term in the left part of relation (9) should be corrected to .

Response:  Thanks, we apologize for my carelessness. We have added spaces between formulas and commas and corrected the errors in expressions (9) and (10) with red mark.

In line 127 the term “oneand” should be corrected also as the term “usedto” in line 157, term “distributionon” in line 189 and term “ofmixture” in line 245.

Response:  Thanks, we feel sorry for our poor writings. We have added spaces between words.

The authors present all the quantities associated with the flow as average values and pulsationsand construct the equation (3). But such representation is not used for temperature. Why? What values of ui and T (average, pulsation) are included in the equation (9)? Why are the equations for pulsations of flow, temperature and stress not given and why is it not necessary to solve these equations? What values are shown in Figs. 4, 5, 7, 8, 10? Is it the average value or the average plus the pulsation for each quantity? What, then, can be said about temperature?

Response:  Thanks, we are very sorry for not being able to express the numerical values in the formula clearly. In this article, the RANS method is used to study turbulence. The velocity and temperature in the calculation are average values, not pulsation values. Equations 6, 7, 8, 9, 11 have been modified and Equation 10 has been added to further explain how to solve for temperature. The speeds in the subsequent analysis are average speed and average temperature.

We have added this part in lines 135, 136.

What is g in (11)? Is this the acceleration of gravity?

Response:  Thanks, g is the acceleration of gravity. We have added a note below Equation 11.

It is completely incomprehensible from 2 how are arranged the lines 1, 2, and so on, and planes 1, 2, and so on? This Figure requires considerable revision. It is necessary to think carefully about how to show the reader these lines and surfaces most clearly and to explain in the paper text the location of these elements and why the authors show them.

Response:  Thanks, we have redrawn Fig. 2 and highlighted the lines and sections in the figure 3. It can be seen from Fig. 6 that the iron flow velocity at the bottom of the main trough and the lower part of the side wall surface is large, so the analysis of the shear stress distribution of line 1 (the bottom edge of the center plane) and line 2 (Pass the line 3 to make the parallel surface at the bottom surface of the main trough, the intersection of the parallel surface and the coupling front surface) is analyzed.

The wall of the slag molten iron contacting the refractory has a bottom surface and two sides. Since slag port was ignored in this study, the front and back side of the model are symmetrical. So in this paper, only need to consider the change of half of the main trough. The selection of plane 1 and plane 2 is sufficient to show the temperature change characteristics within the range of the model size. Therefore, the interval between the center plane and plane 1 and plane 2 is 0.2 m, and between plane 3 and plane 2 is 0.1 m since the temperature change at the wall is more obvious. We have added this part in lines 174-184, 248, 249.

The equation under number (11) on page 7 should have number (12). What do the authors understand in this equation by the operation of dividing a vector by a vector: / ?

Response:  Thanks, we feel so sorry for our carelessness.  /  is the velocity gradient on perpendicular to the direction of fluid, the rate of change of velocity perpendicular to the direction of fluid movement, the unit is s-1. The above adds after Equation 12.

What is the residual error, the initial residual curve? This needs to be explained in the paper text.

Response:  Thanks, the residuals include internal residuals and external residuals. When performing non-steady state calculations in openfoam, the number of iterations in each time step is determined by the internal residuals. When the convergence criterion is reached, the next time step is performed for calculation. The external residual (initial residual) is the difference between the calculated results at adjacent time steps. The smaller the residual, the better the result. We have added this part in lines 188-193.

In Figs. 4, 7 and 10 you must specify the horizontal dimensions in meters so that reader can set the position of each cross-section. It is also necessary to explain why in some places of the lower and left-side trough surfaces the flow velocities are directed not tangentially to these surfaces. Why do these speeds not correspond to the specified boundary conditions and are not equal to zero?

Response:  Thanks, we have specified the horizontal dimensions in meters in Fig.4, 7 and 10.

Because those places are not the outermost grid of flow in main trough . We have checked the outermost flow velocities are directed tangentially to related surfaces. The internal flow is affected by turbulence, so exists the lower and left-side trough surfaces the flow velocities are directed not tangentially to these surfaces.

Because only the velocity distribution of the internal field (liquid) is selected during post-processing data analysizing. If the boundary layer is selected, zero velocity at the boundary will also appear on the graph,and too many arrows interfere with the analysis of the internal velocity distribution。

The location of the cross-section, shown in Figs. 5 and 9, in trough is unclear.

Response:  Thanks, we have redrawn Fig. 2 and highlighted the lines and sections in the Fig. 3.

Round 2

Reviewer 1 Report

Dear Authors

Thank you for your kind revision

The most of the reviewers comments were addressed

I have still some minor revisions to pose to your attention

1) I have a doubt about the service life of the main through with respect the service life of the blast furnace. If the through must be maintained every 35 days and in the whole service life of the BF you need at least 5 repairs, it means that the BF service life is only 175 days. I think this is not reasonable. Could you check and correct your statement?

2) In discussion and in conclusions sections a reference to which are the best conditions of tapping are missing. Please add a discussion about the best tapping angle to reduce the wear of the through and try to estimate the increase in service life of the through with respect the literature data already shown in the introduction

Best Regards

Author Response

Dear Editors,

   Thanks you for informing us the positive feedbacks; as well as the reviewers for quickly processing reviews of our manuscript. We feel very thanks for your professional review work on our article. We have made extensive corrections to our previous draft, the detailed corrections are listed below.

Reviewer #1:

1) I have a doubt about the service life of the main through with respect the service life of the blast furnace. If the through must be maintained every 35 days and in the whole service life of the BF you need at least 5 repairs, it means that the BF service life is only 175 days. I think this is not reasonable. Could you check and correct your statement?

Response:  Thanks for your suggestion.

The full service life of the blast furnace = Each service life (35 days) * Number of service life of main trough per year (9 to10).

The original text was inappropriate and we modified the description in the original text. “Due to the harsh working environment, main trough of a 3000 m3 blast furnace has 9 to 10 campaigns per year and 45 tons of casting material (price 857 $ per ton) is needed for each campaign. Each campaign is about 35 days and needs five minor maintainances. Each minor rmaintainance consumes 3 tons of ramming material (price 823 $ per ton).” We have changed this part in lines 30-33.

2) In discussion and in conclusions sections a reference to which are the best conditions of tapping are missing. Please add a discussion about the best tapping angle to reduce the wear of the through and try to estimate the increase in service life of the through with respect the literature data already shown in the introduction

Response: Thanks for your suggestion. The taphole angle of blast furnace has different vaules in different stages of the furnace. The blast furnace in this study is at a stage where the furnace temperature is stable, and the taphole angle range is 7 to 12°. According to our research, as the tapping angle increases within this range, the wall shear stress increases. When the taphole angle is 7 °, mechanical erosion of the trough has the smallest values. Which increases the strength of the scour refracotry material, the main scouring position of the main trough and the position of the high temperature area will move downstream. During the tapping, reasonable adjustment of the taphole angle can reduce the extent of mechanical erosion in some areas of the main trough, which is helpful to improve the utilization efficiency of the main trough.

The campaign of a main trough is 35 days shown in the introduction, and the taphole angle is 10°. Compared with the normal taphole angle (10°), the maximum wall shear stress is reduced by 16%, when the taphole angle is 7°. Futhermore the campaign of the main trough is estimated to increase by 5 days.

We have added this part in lines 316, 317, 322, 323, 326, 327, 334-337.

Reviewer 3 Report

The authors found it possible to agree with all the comments made in the review, and took them into account in the revised version of the article. I recommend this version for publication.

Author Response

Dear editor,

   Thanks you for informing us the positive feedbacks; as well as the reviewers for quickly processing reviews of our manuscript. We feel very thanks for your professional review work on our article. We decided to keep the original author.

Best wishes.

Kind Regards,

Geyao
